# The Experience of Patients in Chronic Care Management: Applications in Health Technology Assessment (HTA) and Value for Public Health

**DOI:** 10.3390/ijerph19169868

**Published:** 2022-08-10

**Authors:** Federico Pennestrì, Giuseppe Banfi

**Affiliations:** 1Scientific Direction, IRCCS Istituto Ortopedico Galeazzi, 20161 Milan, Italy; 2Scientific Direction, Università Vita-Salute San Raffaele, 20132 Milan, Italy

**Keywords:** patient-reported experience, patient-reported outcomes, health technology assessment, chronic care management, value-based care

## Abstract

Frail chronic patients consume the largest share of resources in advanced healthcare systems, with more hospitals waiting to receive them in the acute phase (awaiting paradigm) than there are effective public health interventions to keep them out of hospitals as much as possible. Effective chronic care management (CCM) requires organizational research as much as biomedical research (and, in some cases, perhaps more). Otherwise, excellent clinical care is wasted by poor coordination among professionals and institutions, with frail patients and their families paying the most expensive price. Comprehensive health technology assessment (HTA) procedures include organizational, social, and ethical dimensions to precisely capture the environmental factors that make medical interventions effective, accessible, and sustainable. Clinical outcomes and financial data are used extensively to evaluate care pathways from the providers’ perspective, but much remains to be done to capture equally important indicators from the perspective of patients and society. The authors hypothesize that the ordinary use of patient-reported experience measurement (PREMs) in HTA can help reduce gaps and inequalities by identifying frail patients on time, curbing the risks of isolation and the burden on care givers, preventing complications and inappropriate emergency care use, improving adherence, health communication and behavior, supporting risk assessment, and relieving the frequency of the healthcare environment.

## 1. Introduction

Patient experience can be understood in two main ways [1,2] (Figure 1). In a broad, phenomenological sense, the experience of patients is the first-person perception of their health conditions. Examples are the experience of sickness, the experience of one’s own body and (dis)ability, the experience of psychosocial (dis)function and the experience of health, which is more easily understood after experiencing a disease.

In a narrow sense, the experience of patients represents the way they perceive the healthcare environment, the care pathways they undergo, and the professionals they encounter. Examples are the access to treatment (the timing of treatment, the distances to be covered, the presence of environmental barriers, the inclusiveness of technologies, treatments, and healthcare professionals), the continuity of care (the health journey across multiple facilities and providers, the support received by professionals within the transitions, the financial sustainability of out-of-pocket treatments, the coordination of individuals and institutions, the amount of bureaucracy to deal with) and the communication with care providers (the clarity of information received, the availability of professionals, the possibility to raise questions and get answers appropriate to the age and cognition of patients).

Both types of experiences significantly affect.
Patient adherence (ability to understand medical advice, actively participate to decisions, comply with prescriptions);Caregiver burden (ability to support the patient and maintain one’s own job, societal role, responsibilities and wellbeing);Quality, safety, and cost-effectiveness of care (avoiding complications, duplicative visits, pharmacological overtreatment, inappropriate care, waiting lists extension, emergency care abuse).

Therefore, capturing the experience of patients can help design different type of interventions to improve individual wellbeing, community health, and financial sustainability (i.e., organizational, educational, financial).

This is even more true in elderly patients affected by psychosocial frailty and chronic morbidities. Non-communicable chronic diseases are known to represent one of the main epidemiological, social, and financial challenges worldwide [3,4,5], with elderly patients and their families struggling to receive the support they need [6]. Much less is known about how to help these patients maintain an acceptable quality of life beyond medical treatments, for instance, by improving the accessibility and coordination of care, tackling isolation in advance and prescribing societal interventions [6,7,8]. When full health is out of reach, the way health care is organized is key for making a pathway sustainable for patients and families, and patient engagement is key for making health care effective, both improving therapeutic adherence and preventive behaviors. Various quantitative studies have demonstrated this [9,10,11].

Frail chronic patients are the population minority consuming the largest share of resources in advanced healthcare systems [6,12,13], with more hospitals waiting to receive them in the acute phase (awaiting paradigm) than effective public health interventions to keep them out of hospitals as much as possible (proactive paradigm). Effective chronic care management (CCM) requires organizational research as much as biomedical research (and, in some cases, perhaps more). Otherwise, excellent clinical care is wasted by poor coordination among professionals and institutions, with frail elderly patients paying the most expensive price [6,14].

Comprehensive Health Technology Assessment (HTA) procedures such as the European Network for Health Technology Assessment (EuNetHTA) Core Model^®^ include organizational, social and ethical dimensions to capture the environmental factors that make medicine more effective, accessible, and sustainable [15,16]. Clinical outcomes and financial data are used extensively to evaluate care pathways from the providers’ perspective (i.e., efficacy, effectiveness, cost-effectiveness), but much remains to be done to capture equally important indicators from the perspective of patients and society [17]. Despite the fact that the importance of ethical and qualitative dimensions in the assessment of healthcare technologies has long been recognized [18,19], the ethical, social, and organizational implications of innovation are still largely underdeveloped, including those areas where they have the greatest potential, such as telemedicine and CCM [20]. In this communication we hypothesize that the ordinary use of patient-reported experience measures (PREMs) can contribute to fill these gaps by
Supporting the ordinary work of chronic care managers, case managers and clinical managers;Focusing on the connections between units (i.e., single professionals, wards and facilities) in addition to their single performances;Enhancing the benefits of clinical care with organizational and educational interventions;Providing specific outcomes for such interventions;Identifying the need for complementary interventions (i.e., social prescribing);Assessing their impact on the care pathway;Evaluating the impact and perception of innovative care technologies aimed specifically at the chronic patient.

## 2. Capturing the Patient Experience: Current Use and Further Opportunities

Increasing questionnaires are developed by researchers to evaluate the healthcare benefits actually perceived by patients. Patient-reported outcome measures (PROMs) are administered directly at patients or proxies at different stages before intervention (when possible) and after (when appropriate), investigating the variations they perceived in relevant domains to their health and health-related quality of life (i.e., pain, fatigue, physical function, psychosocial function, emotional balance, ability to perform the activities of day-living). As such, they mainly describe the patient experience in the broad sense. PROMs can be employed in HTA to integrate conventional clinical outcomes (physician-reported) and financial data: indeed, plenty of questionnaires are available to grasp different conditions (i.e., overall wellbeing, condition-specific, organ-specific, patient-specific) and support care decisions on different levels (clinical comparisons, resources allocation, and performance remuneration) [21,22]. On the contrary, PREM questionnaires describe the patient experience in the narrow sense, as they are more focused on evaluating the way treatments are administered (i.e., context, relations, availability of professionals) rather than the health benefits of treatments themselves. Some interesting items of investigation are, for example [23,24,25,26,27]:Waiting time to receive care;Discharge information received;Support received during access to care, treatments and follow-up;Perceived degree of communication and cooperation from the team taking care of the patient;Safety concerns;Relations with the staff;Awareness on part of the medical practitioner about a patient admitted to hospital and/or undergoing surgery;Involvement in care pathway decisions;Being treated in an age-proper way (i.e., pediatric and geriatric patients);Being encouraged to ask;Being listened carefully;Clarity of information received;Kindness and courtesy;Adequacy of the healthcare environment (i.e., silence);Possibility to ask questions before subscribing informed consent.

Similar to PROMs, these questionnaires can be administered at different stages of care, directly to the patient or proxy (i.e., parents of pediatric patients, adult children of geriatric patients, and care givers in general). Unlike PROMs, PREM questionnaires are more spread as bottom-up, spot initiatives for the internal use of facilities or occasional comparisons between facilities [24,26,27]. 

Most importantly, PREMs are more often employed within the single facilities than *across* the cycle of care. While the former use can help enhance the individual performance of professionals and units (i.e., monitoring safety and improving soft-skills) [21,26,27,28], the latter use could significantly enhance the management of chronic patients enrolled in long term multidisciplinary care programs. Solid coordination among professionals and easy connections among facilities make a real difference in the everyday life of these patients and their families in terms of reduced disorientation and distress, more appropriate care prescriptions, reduced burden on caregivers, reduced complications and acute care admission rates (personal value). Therefore, the ordinary use of PREMs in HTA can also benefit society and public health more in general, increasing the (cost-)effectiveness of conventional care pathways (technical value), reducing inappropriate care volumes and waiting lists (allocative value), and maintaining productivity in adult, non-elderly patients (societal value) [29].

On these grounds, PREMs seem the most suitable indicators to identify chronic care gaps directly from the patients, and tackle clinical, organizational, and financial waste across the continuity of care [30].

## 3. Potential Benefits of PREMs in CCM

Although there is no universal protocol for CCM, the care of patients with chronic (co)morbidities generally requires the design of a patient-oriented pathway around integrated, multidisciplinary providers (social care eventually included), takes an indefinite duration, is subject to updates and may be binding for patients. To meet these requirements, three fundamental pillars are commonly introduced:The direction of a clinical manager, be it the general practitioner for monopathological, non-complex patients or a specialist physician with expertise on the prevailing condition in more critical, frequently hospitalized patients, ensuring the coordination of multiple treatments from a medical perspective;The coordination of a case manager, be a nurse or another non-medical care professional (i.e., the physical therapist for musculoskeletal conditions) who works as the reference point for the patient and eventual care giver, booking visits, prescribing drugs, renovating routine referrals, monitoring ordinary parameters, ensuring the connection of units from an organizational perspective (including, for example, the provision of social support);The provision of all necessary treatments from a care manager, either directly or indirectly (i.e., purchasing services from other providers). The more gravity and complexity of the patient, the greater the possibility that the care manager is part of a hospital network; the lower the severity and complexity, the greater the possibility that the care manager is a primary care provider (i.e., a general practitioners outpatient clinic). If clinical managers and case managers are flesh-and-bone professionals, the care manager represents the formal institution they work for.

Pay-per-coordination solutions are often introduced to encourage appropriate CCM. For instance, the Chronic Related Groups experience (CreG) in the Region of Lombardy, Northern Italy, introduced a bundle payment covering all of the treatments needed by patients affected by some pilot chronic diseases (diabetes, heart failure, chronic-obstructive pulmonary disease, hypertension), reducing the use of emergency care and hospital admissions given equal health conditions perceived by patients [8]. That pilot experience was extended in a more comprehensive policy reform including drugs, outpatient visits, hospitalization and social care in the chronic patient budget, although social care indicators still need to be developed [8]. 

PREMs can help policy makers fill this gap, both when social care is functional to health care adherence (i.e., taking patients to healthcare facilities or helping them comply with therapy) and when social care is aimed at improving general wellbeing and safety. Personal Health Budgets for chronic patients and their care givers were also introduced in the United Kingdom under the general framework of universal personalized care, coordinated by clinical managers working for NHS care managers and subsidiary providers [31]. 

In both programs the collection of PREMs can help policy makers, clinical managers and care managers identify gaps through the cycle of care and adopt cost-effective solutions *in support* of conventional treatments (i.e., re-ablement, restorative care, social prescribing), investing to make bureaucracy more effective and integrating different databases on shared digital platforms (i.e., to obtain timely exemptions from out-of-pocket expenditure for life-sustaining ordinary treatment). In the NHS, social prescribing is even being recommended *in alternative* of conventional medical treatment where it is cost-effective and appropriate [32]. PREMs not only point out the flaws to be addressed by complementary, organizational arrangements, but they can also be employed to assess their effectiveness. 

The ordinary issues faced by frail, chronic patients in advanced healthcare systems make the systematic use of PREMs in CCM worth-trying, no matter how excellent is the quality of care provided in acute facilities. In Sweden, a study on elderly patients hospitalized for myocardial infarction reports the consequences of health care fragmentation before and after the admission to hospital: 50% were not visited by any operator up to one month before emergency hospitalization; 79% has resorted to the emergency room for inappropriate needs; 14% had not received any written information to understand who to contact from the moment of discharge; 30% did not know who to consult in case of adverse events and complications) [33]. The Region of Lombardy is characterized by one of the greatest hospital networks in the entire Italy, but struggles to bear the pressure of 30% chronic patients consuming 70% healthcare resources, as consequence of poor coordination between primary and secondary care, further aggravated by being at the center of the COVID-19 pandemic [34]. Isolated patients worldwide are exposed to preventable mortality up to 50% more than patients who have someone close to rely on [35]: “an effect on mortality comparable to smoking 15 cigarettes a day, higher than obesity and chronic physical inactivity” [36]. On similar grounds, Van Oppen and coauthors [37] argue that PREMs should also be collected in emergency care facilities, once the patient condition has been stabilized and specific questionnaires have been identified and validated: indeed, chronic frail patients often abuse emergency care services as a consequence of isolation and disorientation. Healthcare fragmentation is a thorny health and sustainability problem, especially for those patients who struggle most to receive essential ordinary care, turning the inverse care law a problem of both universal and market-driven healthcare systems [38,39]. 

Collecting PREMs under the provision of managed (chronic) care can help reduce gaps and inequalities by identifying frail patients on time, reducing the risks of isolation and the burden on care givers, preventing complications and inappropriate emergency care use, improving adherence, health communication and behavior, supporting risk assessment, and relieving the frequency of the healthcare environment whenever possible (consider the additional benefits in terms of reducing the spread of infections and preserving public health). Clinical managers should explain the pathway in a plain language and be open to questions. Case managers should spare avoidable visits and contacts to clinical managers by providing routine information, ordinary prescriptions and low-complexity parameters monitoring, possibly taking advantage of remote care technology. Telemedicine supported by artificial intelligence for instance may help address the shortage of specialized care professionals outside hospitals and major urban centers. Care managers should identify room for improvement across the organization taking advantage of the direct perspective of patient and care giver, enhancing the soft-skills of health care professionals through financial incentives and/or professional training and/or making the whole service more accessible (i.e., removing environmental barriers, placing more benches or designing intra-hospital pathways to help patients and care givers).

CCM does not require the administration of additional treatments; it works on the way these treatments are integrated in a patient-centered perspective (that is, in a way that can actually benefit different patients with different needs, resources, and characteristics). From this perspective, PREMs are optimal indicators for organizational, legal, social, and ethical impact in HTA procedures. For example, the following items were employed to express the equity, legal, and organizational dimensions of an enhanced-recovery after surgery perioperative pathway [40]:

Equity domain:Accessibility of the technology in the area (proximity of adequate rehabilitation facilities to the hospital where surgery was performed);Accessibility of the treatment to protected citizens, such as frail elderly and disabled patients;Accessibility of the technology to all patient categories, including those affected by severe comorbidities;Accessibility to post-discharge facilities;Post-discharge facility decision is shared with the patient;Potential impact in reducing waiting lists;General inclusivity;The technology respects the cultural, moral and religious identity of the patient.

Burden removed by the technology:Ability to safeguard the patient’s autonomy;Economic accessibility to the patient;Removal of the disease-related social costs in charge of the patient;(Impact of) Social determinants of compliance and technology comprehension;Patient satisfaction;Ability to improve the patient’s quality of life (of the patient);Ability to improve function (perceived);Ability to improve the family caregiver’s quality of life;Reduced time spent in the healthcare environment.

Legal domain:Need to regulate the technology (local or national guidelines);Regulatory protection of specific patients;Is patient-information about the technology exhaustive?Degree of sensitive data protection;Safety criteria.

Most of these items would be appropriate to assess a CCM program and many PREMs among those exemplified in the previous section would be suitable outcomes. In this study, the items were assessed by qualitative interviews to the healthcare professionals involved in the pathway. The use of standard PREMs would bring the assessment to a further level, taking into consideration the direct experience of patients and caregivers and allowing for comparisons between different patients, professionals, and facilities.

## 4. Limitations

Patient-reported experience should not be confused with patient-satisfaction, although the latter may be included as a single indicator among others [21]. Moreover, patient-satisfaction should not be confused with customer satisfaction, which may be related to other environmental variables not concerning the health of patients (i.e., the beauty of outpatient accommodations or the extensive selection of the hospital menu), or not substantially. However, any patient-reported indicator (either PROMs and PREMs) is subject by definition to the expectations of patients [21], which may blur to some extent any reasonable distinction between what falls under the domain of healthcare professionals (i.e., the ability to speak plainly and more kindly to patients and families) and the rush timetables they have to meet; what falls under the treatments prescribed and what falls under the patient’s own behavior and commitment; and what ultimately falls under the domain of medicine and what not (i.e., the full recovery of chronic patients or the perfectionist attitudes of healthism and consumerism). The authors believe this is not a reason to disinvest on the collection of PREMs, but rather that this is a reason to standardize and spread their use to absorb excessive expectations within the average of large-scale patients.

## 5. Conclusions

We believe that the ordinary use of PREMs in HTA can increase the direct (patient-related) and indirect (care giver- and society-related) benefits of CCM by enhancing all of the dimensions of value: personal value, by improving or maintaining the wellbeing of patients; technical value, by improving the (cost-)effectiveness of medical treatment through better adherence and patient-centered organization; allocative value, by reducing waste and freeing up financial, human and time resources; and societal value, by translating health benefits into socio-economic benefits, such as more productivity, reduced welfare dependency, and more time to dedicate to those people and activities which make everyday life worthy-living.

## Figures and Tables

**Figure 1 ijerph-19-09868-f001:**
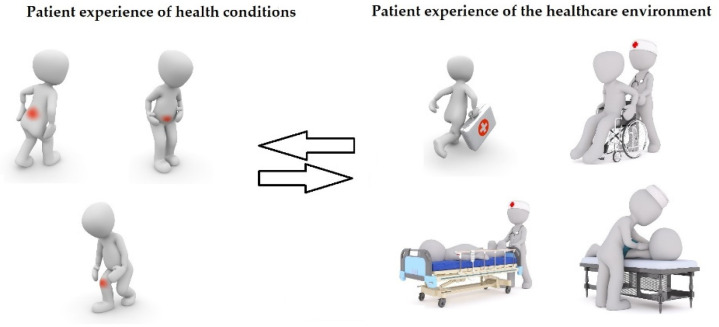
Patient experiences in a broad and a narrow sense.

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
