# Peer review of "The Experience of Patients in Chronic Care Management: Applications in Health Technology Assessment (HTA) and Value for Public Health"

_ijerph, 2022, doi:10.3390/ijerph19169868_

Round 1

Reviewer 1 Report

Thank you for giving me the opportunity to read this communication. It is a summary of existing knowledge on the subject and does not add anything new to the scientific community. I suggest to the authors that they advance their hypothesis. Thank you

Author Response

Dear Editor,

Dear Reviewers,

Thanks for your recommendations, which we did our best to meet.

Please find our point-by-point reply in blue.

Kind regards,

the authors.

Reviewer 1

Thank you for giving me the opportunity to read this communication.

It is a summary of existing knowledge on the subject and does not add anything new to the scientific community.

I suggest to the authors that they advance their hypothesis. Thank you

Thanks for your suggestion.

We were not probably clear enough in advancing our hypothesis and made our best to support it further.

Papers dealing with the use of PREMs in specific chronic care assessments do exist, and they are not all mentioned in our communication. Our aim was not really to summarize the existing literature on the topic, but to argue for their ordinary use in formal, comprehensive HTA procedures, bringing potential applications to light, as their current use is limited and spotted (i.e., bottom-up) at best. We provide examples of their use in chronic care management in ways that we did not find in the literature, from supporting the work of chronic care managers, care managers and clinical managers to introducing specific outcomes for the evaluation of organizational and educational interventions. We explicitly promote their use in order to fill some HTA gaps (i.e., qualitative assessment, social, ethical, organizational dimensions) as recognized by experts, policy makers and scholars, including relevant applications for chronic patients (i.e., remote ordinary care and telemedicine). We added lines 99-118 to highlight these details and advance the general hypothesis more clearly.
Please do not hesitate to ask for further clarifications.

Thank you.

Reviewer 2 Report

This is a Communication paper on the Patient-Reported Experience Measurement (PREMs) and its applications in Health Technology Assessment (HTA) and value for public health.

The article addresses a relevant topic of great interest and it is well written.

I have minor comments and a few suggestions.

Introduction

- Although in general the article is easy to read, perhaps a schematic figure could help to understand more easily the explanation of the Introduction on the patient experience and the two main ways. The authors could consider including such a figure, although it is not essential.

- In the first part of the Introduction, lines 28 to 54, there are no bibliographical references. I wonder if there is no literature that has mentioned this before.

-The reference #14 is a long document, 275 pages, so it would be convenient if the reference were to the relevant Chapter on this topic instead of the entire document.

-Does reference #15 correspond to a document? In that case, the authors should complete the data of the same (title, publisher, year of publication, etc.). The data provided in the list of references is insufficient for the reader to find that document. In the text, the authors mention "(p. 172)", I assume it is the page. Instead of mentioning the page in the text, the reference could be that of the chapter.

Capturing the patient experience: current use and further opportunities

-Authors should correct the spelling in “appropricate”, line 131.

-The authors say “At the moment of writing, the string “HTA AND PREMs” does not give results on MEDLINE/PubMed and one result on Embase” (lines 141-143). They could mention, in addition to that string that uses abbreviations, the string using the complete words. I suggest this because it is likely to find a smaller number of results by using abbreviations.

-Please check the references, from 27 to 37. The one that appears as 28 in the text is 27 in the reference list....the one that appears as 38 is 37...etc.

Potential benefits of PREMs in CCM

Lines 174-175, please check the phrase "reducing the use of emergency care and hospital ad-174 missions equal health conditions perceived by patients " which is not clear.

Limitations

Reference 17, in the two places where it reads “[17] (p. 5)”, it is not necessary to mention the article page.

Author Response

Dear Editor,
Dear Reviewers,
Thanks for your recommendations, which we did our best to meet.
Please find our point-by-point reply in blue.
Kind regards,
the authors.
Reviewer 2
This is a Communication paper on the Patient-Reported Experience Measurement (PREMs) and its applications in Health Technology Assessment (HTA) and value for public health. The article addresses a relevant topic of great interest and it is well written.
I have minor comments and a few suggestions.
Introduction
- Although in general the article is easy to read, perhaps a schematic figure could help to understand more easily the explanation of the Introduction on the patient experience and the two main ways. The authors could consider including such a figure, although it is not essential.
Thank you for all the recommendations.
We included a figure to represent the two main ways in which patient experience is discussed in the paper.
- In the first part of the Introduction, lines 28 to 54, there are no bibliographical references. I wonder if there is no literature that has mentioned this before.
We added two bibliographical references which refer to both types of patient experiences, their meaning and main applications. More references from the literature were gradually included in the paper.
- The reference #14 is a long document, 275 pages, so it would be convenient if the reference were to the relevant Chapter on this topic instead of the entire document.
We made reference to the relevant Chapter.
- Does reference #15 correspond to a document? In that case, the authors should complete the data of the same (title, publisher, year of publication, etc.). The data provided in the list of references is insufficient for the reader to find that document. In the text, the authors mention "(p. 172)", I assume it is the page. Instead of mentioning the page in the text, the reference could be that of the chapter.
We corrected the reference that was actually wrong.
Capturing the patient experience: current use and further opportunities
- Authors should correct the spelling in “appropricate”, line 131.
We corrected the spelling.
- The authors say “At the moment of writing, the string “HTA AND PREMs” does not give results on MEDLINE/PubMed and one result on Embase” (lines 141-143). They could mention, in addition to that string that uses abbreviations, the string using the complete words. I suggest this because it is likely to find a smaller number of results by using abbreviations.
This is true, we ran the same string with the extended keywords on PubMed and Embase and found more results. Then we had a look on some of the papers resulting from these attempts, in order to capture whether they were relevant to our purpose. We saw that “assessment” was used many times without any relationship with formal HTA procedures, and “experience” was used many times without any relationship with the formal collection of PREM questionnaires. We could not provide a systematic analysis of these results, because it would have become entirely another paper. Therefore, although some papers on the use of PREMs in HTA procedures may exist, we still believe that their use is limited and spotted, and made reference to three additional papers (18-20) in support of our hypothesis, which we further highlighted in detail between line 99 and 118.  We finally removed the statement about the total lack of studies of this type.
- Please check the references, from 27 to 37. The one that appears as 28 in the text is 27 in the reference list.... the one that appears as 38 is 37...etc.
We entirely revised the bibliography, moving references where appropriate, removing two references and adding five references.
Potential benefits of PREMs in CCM
- Lines 174-175, please check the phrase "reducing the use of emergency care and hospital ad-174 missions equal health conditions perceived by patients " which is not clear.
We checked and corrected the phrase, which we hope now is clearer. Given the same health conditions, emergency care and hospital admissions were reduced after the pilot experience, which may be interpreted as a positive outcome both in organizational and clinical terms (i.e., reduced hospital attendance and increased appropriateness).
Limitations
- Reference 17, in the two places where it reads “[17] (p. 5)”, it is not necessary to mention the article page.
We removed the article page both times.
We hope we have met your recommendations. Please do not hesitate to ask for further clarifications. Thank you.

Reviewer 3 Report

The present paper enhances the importance of widespread use of Patient Reported Experience Measures in Health Technology Assessment and argues that can increase the direct and indirect benefits of chronic care management by assessing a few prior published research.

As it seems this issue has a high level of novelty (supported by a lack of available bibliography related) could be interesting to share this paper with the scientific community. Nevertheless, I suggest editing the paper and submitting it as a Letter to Editor because currently it seems a little redundant and has not had the empiric base required in an original report

Author Response

Dear Editor,
Dear Reviewers,
Thanks for your recommendations, which we did our best to meet.
Please find our point-by-point reply in blue.
Kind regards,
the authors.
Reviewer 3
The present paper enhances the importance of widespread use of Patient Reported Experience Measures in Health Technology Assessment and argues that can increase the direct and indirect benefits of chronic care management by assessing a few prior published research. As it seems this issue has a high level of novelty (supported by a lack of available bibliography related) could be interesting to share this paper with the scientific community. Nevertheless, I suggest editing the paper and submitting it as a Letter to Editor because currently it seems a little redundant and has not had the empiric base required in an original report.
Thanks for your comments and recommendation.
We are conscious that our paper can not be considered as original research, in fact we have submitted it as a communication, after agreement with one of the Special Issue Editor.
Please do not hesitate to ask for further clarifications. Thank you.

Round 2

Reviewer 1 Report

Thank you very much for listenting to my suggestions.

Reviewer 3 Report

I agree to accept this reviewed version for publication